# Toward Inferring Potts Models for Phylogenetically Correlated Sequence Data

**DOI:** 10.3390/e21111090

**Published:** 2019-11-07

**Authors:** Edwin Rodriguez Horta, Pierre Barrat-Charlaix, Martin Weigt

**Affiliations:** 1Laboratoire de Biologie Computationnelle et Quantitative (LCQB), Institut de Biologie Paris-Seine, Sorbonne Université, Centre national de la recherche scientifique (CNRS), 75005 Paris, France; erodriguezh1990@gmail.com (E.R.H.); p.barrat@live.fr (P.B.-C.); 2Group of Complex Systems and Statistical Physics, Department of Theoretical Physics, Physics Faculty, University of Havana, La Habana 10400, Cuba; 3Biozentrum, University of Basel, 4056 Basel, Switzerland

**Keywords:** phylogeny, co-evolution, direct coupling analysis

## Abstract

Global coevolutionary models of protein families have become increasingly popular due to their capacity to predict residue–residue contacts from sequence information, but also to predict fitness effects of amino acid substitutions or to infer protein–protein interactions. The central idea in these models is to construct a probability distribution, a Potts model, that reproduces single and pairwise frequencies of amino acids found in natural sequences of the protein family. This approach treats sequences from the family as independent samples, completely ignoring phylogenetic relations between them. This simplification is known to lead to potentially biased estimates of the parameters of the model, decreasing their biological relevance. Current workarounds for this problem, such as reweighting sequences, are poorly understood and not principled. Here, we propose an inference scheme that takes the phylogeny of a protein family into account in order to correct biases in estimating the frequencies of amino acids. Using artificial data, we show that a Potts model inferred using these corrected frequencies performs better in predicting contacts and fitness effect of mutations. First, only partially successful tests on real protein data are presented, too.

## 1. Introduction

Based on the rapidly growing availability of biological sequence data [1,2,3], statistical models of sequences have gained considerable interest over the last years [4,5,6,7]. In this context, the direct coupling analysis (DCA) [8] takes inspiration from inverse statistical physics [9]: it aims at describing the sequence variability of sets of evolutionarily related protein sequences—so-called homologous protein families—via Potts models. Such a model gives a probability
(1)$$P\left(\underline{A}\right)=\frac{1}{Z}\exp\left\{\sum_{1\leq i<j\leq L}J_{ij}\left(A_{i},A_{j}\right)+\sum_{1\leq i\leq L}h_{i}\left(A_{i}\right)\right\}$$
to each aligned amino acid sequence $\underline{A}=\left(A_{1},\ldots ,A_{L}\right)$ of length *L*, with the $A_{i}\in A=\left\{A,C,\ldots ,Y,-\right\}$ being either one of the 20 amino acids, or an alignment gap, “–”, representing amino acid insertions or deletions. The total alphabet size is q=|A|=21. Strong statistical couplings $J_{ij}$ between different positions have been found to be indicative of contacts of the corresponding amino acids in the three-dimensional protein fold, thereby facilitating protein structure prediction from sequence information [10,11]. Furthermore, the statistical energy landscape (i.e., the Hamiltonian $H(\underline{A})=-\sum_{1\leq i<j\leq L}J_{ij}\left(A_{i},A_{j}\right)-\sum_{1\leq i\leq L}h_{i}\left(A_{i}\right)$ of the Potts model in Equation (Equation 1)) around a sequence has been found to be informative about the effects of mutations on the protein’s functionality (or fitness) [12].

Sequence data for protein families are typically available as multiple-sequence alignments (MSA), i.e., as collections ${\left\{{\underline{A}}^{m}\right\}}_{m=1\ldots M}$ of *M* distinct sequences of the same (aligned) length *L*. To fit the model $P(\underline{A})$ in Equation (Equation 1) to these data, typically a very strong assumption is made: the MSA is considered an independently and identically distributed sample of the statistical model. This implies that the model can be inferred by maximizing the likelihood
(2)$$L_{i.i.d.}\left(\left\{J_{ij}\left(A,B\right),h_{i}\left(A\right)\right\}\;|\;\left\{{\underline{A}}^{m}\right\}\right)=\prod_{m=1}^{M}P\left({\underline{A}}^{m}\right)$$
over all couplings $J_{ij}\left(A,B\right)$ and fields $h_{i}\left(A\right)$. Although this task is computationally hard—it requires in particular the calculation of the partition function *Z* in Equation (Equation 1) as a sum over $21^{L}$ sequences—numerous approximation schemes have been developed and are reviewed in [7,9].

However, the evolutionary history of proteins is in evident contradiction with the assumption of statistical independence between sequences. The very notion of homologous protein families implies that present sequences derive from a common ancestor. Even if the divergence time from this common ancestor is long enough to result in overall high sequence diversity, some protein sequences may be found in closely related species, or may go back to a relatively recent event of duplication or horizontal gene transfer. This is commonly observable in MSA, where sequences differing by only few amino acids are frequent.

The evolutionary history of a protein family is typically represented by a phylogenetic tree [13], cf. Figure 1 for a simple example. Sequences observable today correspond to the leaves of this tree, and the common ancestor to its root. Branching points correspond to events separating two sequences, typically via speciation, duplication, or horizontal gene transfer. On distinct branches, proteins are assumed to evolve independently.

However, if the branching event separating two sequences, ${\underline{A}}^{1}$ and ${\underline{A}}^{2}$, took place some time Δt in the past, the joint probability should be written as $P({\underline{A}}^{1},{\underline{A}}^{2}|\Delta t)$, which a priori differs from the product of the two equilibrium probabilities. This becomes evident in the case Δt=0, where ${\underline{A}}^{1}={\underline{A}}^{2}$, and thus $P\left({\underline{A}}^{1},{\underline{A}}^{2}|\Delta t=0\right)=P\left({\underline{A}}^{1}\right)\;\delta_{{\underline{A}}^{1},{\underline{A}}^{2}}$ with δ being the multidimensional Kronecker symbol. This extreme situation can be observed in protein families, where, e.g., protein sequences of different strains of the same species differ at most by a few mutations. Note that nevertheless each single sequence may be in equilibrium: $\sum_{{\underline{A}}^{2}}P\left({\underline{A}}^{1},{\underline{A}}^{2}|\Delta t\right)=P\left({\underline{A}}^{1}\right)$ for all Δt, and similarly for ${\underline{A}}^{2}$.

The statistical dependence between homologous proteins poses an important problem to the inference of our statistical model $P(\underline{A})$. The likelihood of the coupling and field parameters given the MSA ${\left\{{\underline{A}}^{m}\right\}}_{m=1\ldots M}$ and the phylogenetic tree T
(3)$$L\left(\left\{J_{ij}\left(A,B\right),h_{i}\left(A\right)\right\}\;|\;\left\{{\underline{A}}^{m}\right\},T\right)\neq L_{i.i.d.}\left(\left\{J_{ij}\left(A,B\right),h_{i}\left(A\right)\right\}\;|\;\left\{{\underline{A}}^{m}\right\}\right)$$
does not factorize into a product of single-sequence probabilities $P({\underline{A}}^{m})$. Using the factorized expression (Equation 2) as an approximation leads to biased statistics, as groups of closely related organisms in the family lead to an over-representation of certain regions of sequence space. Two consequences are illustrated in Figure 1: if we do not consider the tree T, columns 3 (red/blue) and 6 (green/orange) seem to have an equivalent statistics and equal single-site entropies. However, observing the tree, we see that column 3 can be explained by a single mutation in one of the early branches of the tree, whereas column 6 requires at least two mutations in more recent branches. The same amplification of mutations in subtrees may also lead to spurious correlations in the amino acid usage of column pairs. The amino acid usage of columns 3 and 8 (both red/blue) may be explained by a single mutation per site, but suggests a correlation in their joint amino acid usage. It has been recently shown [14] that the phylogenetic bias changes the spectral properties of the correlation matrix. A power-law tail of large eigenvalues emerges from the hierarchical structure the phylogenetic tree, in difference to the Marchenkov–Pastur distribution, which would be present in data lacking both phylogenetic and functional correlations.

Direct inference of a DCA model $P(\underline{A})$, by maximizing the factorized approximation of the likelihood, thus leads to the existence of field and couplings parameters that attempt to model the full biased statistics. As a result, the parameters of the DCA model cannot be expected to accurately represent functional constraints acting on the protein, even if all single sequences were individually distributed according to $P(\underline{A})$.

Usual implementations of DCA [7,8] use the so-called reweighting scheme to account for phylogeny: sequences with more than 80% identity are downweighted, counting for one observation in total. In the Δt=0 case, this has the correct effect of considering ${\underline{A}}^{1}$ and ${\underline{A}}^{2}$ as a single observation. However, in the general setting, this is only a crude correction for the biases, which are generated by the hierarchical sequence organization on the phylogenetic tree.

Here, we aim at designing a more principled method of taking phylogenetic effects explicitly into account. This is done in Section 2, where an approximate but computationally feasible correction of phylogenetic biases is proposed. Section 2.4 discusses how the resulting corrected one- and two-site statistics can be translated into a corrected DCA model. Section 3 shows, first, results on artificial but well-controlled data, which show that our approach is able to correct the statistics of the data, and in turn to improve Potts model inference. Results on real protein data are also shown in this section. The work is concluded with a Discussion in Section 4.

## 2. Methods

Quantitatively, the evolutionary process can be defined by its propagator $P({\underline{A}}^{2}|{\underline{A}}^{1},\Delta t)$: the probability of observing sequence ${\underline{A}}^{2}$ knowing that it has sequence ${\underline{A}}^{1}$ as an ancestor at a time Δt in the past. For the evolutionary process to be stationary, the propagator should satisfy the condition
(4)$$\sum_{{\underline{A}}^{1}}P\left({\underline{A}}^{2}\;|\;{\underline{A}}^{1},\Delta t\right)P\left({\underline{A}}^{1}\right)=P\left({\underline{A}}^{2}\right)\;.$$
The equilibrium distribution of sequences can be recovered by taking Δt→∞, making sequence ${\underline{A}}^{2}$ independent from ${\underline{A}}^{1}$. Knowledge of the propagator and the phylogenetic tree would allow us to calculate the likelihood Equation (Equation 3) using Felsenstein’s pruning algorithm [15]. Note that this model of evolutionary dynamics can easily take into account point mutations, deletions, and insertions, but not large-scale rearrangements like intragenic recombination, which would invalidate the assumption of the existence of a phylogenetic tree. It can take into account selection when the Hamiltonian is considered to be a fitness proxy, but it cannot take into account changes in selection, which would invalidate the assumption of stationary evolution. It can take into account changes in mutation rate when times Δt are measured in terms of a molecular clock rather then in physical time.

Assume the phylogenetic gene tree T to be given, with nodes indexed by *n* (we do not consider the problem of tree inference here). Following the description of Felsenstein’s pruning algorithm in [16], let $L^{n}\left(\underline{A}\right)$ be the conditional probability of observing all existing sequences that share *n* as an ancestor, given that the sequence of this ancestor is $\underline{A}$, but without any information on the sequences at potential intermediary nodes inside the subtree of T rooted in *n*. If *n* itself represents a leaf node, i.e., an existing sequence ${\underline{A}}^{n}$, we trivially have $L^{n}\left(\underline{A}\right)=\delta_{\underline{A},{\underline{A}}^{n}}$. For any internal node of the tree, we find the recursion relation illustrated in Figure 2A:(5)$$L^{n}\left(\underline{A}\right)=\prod_{m\in C(n)}\left[\sum_{\underline{B}}P\left(\underline{B}|\underline{A},\Delta t^{m}\right)L^{m}\left(\underline{B}\right)\right]\;,$$
where C(n) collects the children nodes of *n* and $\Delta t^{m}$ equals the time separating node *m* from its direct ancestor *n*. This recursion can be conducted from the leaves to the root *r* of the tree, with $L^{r}\left(\underline{A}\right)$ as a result. As the sequence of the root of the tree is unknown, it is necessary to sum one more time over all possibilities for this sequence. The probability of observing the sequences of the initial MSA given the tree T and the model parameters, or equivalently the likelihood of the parameters given the MSA and the tree, is given as
(6)$$L\left(\left\{J_{ij}\left(A,B\right),h_{i}\left(A\right)\right\}\;|\;\left\{{\underline{A}}^{m}\right\},T\right)=\sum_{\underline{A}}P\left(\underline{A}\right)L^{r}\left(\underline{A}\right)\;,$$
which obviously differs from the factorized likelihood in Equation (Equation 2). Note that in this last equation we have assumed that the propagator depends on the model parameters, i.e., the couplings $J_{ij}\left(A,B\right)$ and the fields $h_{i}\left(A\right)$. If we would know this dependence explicitly, we might maximize the likelihood in Equation (Equation 6) to infer the equilibrium Potts model Equation (Equation 1) from data.

However, this approach suffers from two major technical problems:The first is that the propagator $P({\underline{A}}^{2}|{\underline{A}}^{1},\Delta t)$ associated to the Potts model is not known a priori. Many distinct microscopic dynamics might lead to the same equilibrium, but the exact evolutionary processes underlying correlated protein evolution are not known. Even if we would assume some dynamics, the propagator for arbitrary time differences Δt would require to sum over all possible evolutionary trajectories going from ${\underline{A}}^{1}$ to ${\underline{A}}^{2}$—but this is intractable in practice.The second problem is that each use of the recursion relation (Equation 5) involves the summation over all possible sequences for each child node of node *n*. This amounts to summing over $21^{L}$ terms each time, with *L* being the sequence length.
Thus, a direct application of this scheme appears impossible for systems of realistic sizes, i.e., for typical sequence lengths L = 50–500. The following sections therefore propose two approximations based on the previously described idea, intending to make the computation of the likelihood tractable.

### 2.1. Approximating Dynamics: Independent-Site Evolution

To reduce the complexity of the problem, we first apply an approximation commonly used in evolutionary biology and phylogeny. The independent-site approximation—also referred to as “single-site” approximation in the following—considers each column of the MSA as evolving independently from all others. In this setting, instead of considering probabilities of observing full sequences as in $L^{n}\left(\underline{A}\right)$, we focus on the distribution of amino acids in one MSA column only. The single-site equivalent of Equation (Equation 5) becomes
(7)$$L_{i}^{n}\left(A\right)=\prod_{m\in C(n)}\left(\sum_{B\in A}P\left(B|A,\Delta t^{m}\right)\;L_{i}^{m}\left(B\right)\right),$$
where $L_{i}^{n}\left(A\right)$ is the probability of observing the state of column *i* in existing sequences that share *n* as an ancestor, given that the sequence of this ancestor contains A∈A at this position. Summations over all possible configurations of internal nodes are replaced by summations over single symbols *B*, resulting in a complexity of O(L×M×q) for computing the *L* sitewise likelihoods. As the number *M* of sequences equals the number of leaves, the number of internal nodes to be summed over equals M−1.

To apply this idea, a propagator is designed using the Felsenstein model of evolution [15] and assuming a constant mutation rate μ (remember that time was measured according to a molecular clock, i.e., the assumption of constant μ is quite natural). In a time interval Δt, no mutations appear, thus with probability $e^{-\mu \Delta t}$, and *B* remains equal to the ancestral amino acid *A*. With probability $\left(1-e^{-\mu \Delta t}\right)$, one or more mutations happen. In this case, the new amino acid at position *i* is assumed to be chosen according to its stationary distribution $P_{i}\left(B\right)=\omega_{i}\left(B\right)$. The following propagator summarizes this process,
(8)$$P_{i}\left(B|A,\Delta t\right)=e^{-\mu \Delta t}\delta_{A,B}+\left(1-e^{-\mu \Delta t}\right)\;\omega_{i}\left(B\right)\;.$$
When using this simple dynamical model and applying the recursion of Equation (Equation 7), it is possible to compute the likelihood of the observed data very efficiently.

The likelihood does not only depend on the MSA and the phylogenetic tree, but also on the value of the mutation rate μ, which in general may be unknown. Within the independent-site approximation, we can easily estimate it using the data. To this aim, we observe that the average of the Hamming distance
(9)$$d_{H}\left(\underline{A},\underline{B}\right)=\sum_{i=1}^{L}\left(1-\delta_{A_{i},B_{i}}\right)$$
between two equilibrium sequences at evolutionary time distance Δt can be easily calculated,
(10)$$\begin{array}{ccc} {\bar{d}}_{H}\left(\Delta t\right) & = & \sum_{i=1}^{L}\sum_{A_{i},B_{i}\in A}\left(1-\delta_{A_{i},B_{i}}\right)\;P_{i}\left(A_{i}|B_{i},\Delta t\right)\;\omega_{i}\left(B_{i}\right)\\ & = & \left(1-e^{-\mu \Delta t}\right)\left[L-\sum_{i,A}\omega_{i}{\left(A\right)}^{2}\right]\\ & = & \left(1-e^{-\mu \Delta t}\right){\bar{d}}_{H}\left(\infty \right)\;. \end{array}$$
Thus, it starts at Hamming distance zero for Δt=0, and approaches exponentially a plateau value, which is given by the average Hamming distance between two independent equilibrium sequences in the independent-site model. In the sequence data, we have no direct observation of parent–child pairs of sequences. The dynamical process given by Equation (Equation 8) is, however, a stationary one satisfying detailed balance $P_{i}\left(A|B,\Delta t\right)\;\omega_{i}\left(B\right)=P_{i}\left(B|A,\Delta t\right)\;\omega_{i}\left(A\right)$. Therefore, we can take any two sequences ${\underline{A}}^{m},{\underline{A}}^{n}$ from the sequence alignment, calculate their Hamming distance together with their time separation on the phylogenetic tree by adding all branch lengths along their connecting path, and use the result as an instance of $d_{H}\left(\Delta t\right)$, cf. Figure 2B. Taking all pairs of sequences from the MSA, we can bin the observed times, calculate average Hamming distances for each time bin, and fit the functional form of Equation (Equation 10) to obtain the desired value of μ, cf. Section 3 for examples.

### 2.2. Approximating Dynamics: Independent-Pair Evolution

Using the independent-site approximation, one recovers the most likely single-site stationary distribution $\omega_{i}\left(A\right)$, given the corresponding MSA column and the topology of the evolutionary tree. Unfortunately, this method is intrinsically unable to correct for spurious correlations such as the one displayed in Figure 1. To reach that aim, we need to find a way to take two-point correlations into account. However, performing phylogenetic analysis with a model of the full sequence is intractable, as is explained at the beginning of this section.

To deal with this dilemma, we choose to use an independent-pair approximation: each pair of sites *i* and *j* is thought of as evolving independently from the others, with a propagator similar to that of Equation (Equation 8). The probability that *i* changes amino acid from *A* to *C* in time Δt, and *j* from *B* to *D*, is defined as
(11)$$\begin{array}{ccc} P_{ij}\left(C,D\;|\;A,B,\Delta t\right) & = & e^{-2\mu \Delta t}\;\delta_{A,C}\;\delta_{B,D}\\ & & +e^{-\mu \Delta t}\left(1-e^{-\mu \Delta t}\right)\left[\omega_{j|i}\left(D\;|\;A\right)\;\delta_{A,C}+\omega_{i|j}\left(C|B\right)\;\delta_{B,D}\right]\\ & & +{\left(1-e^{-\mu \Delta t}\right)}^{2}\;\omega_{ij}\left(C,D\right)\;, \end{array}$$
where $\omega_{ij}\left(C,D\right)$ is the stationary pairwise distribution of sites *i* and *j*, and $\omega_{i|j}\left(C\;|\;B\right)=\omega_{ij}\left(C,B\right)/\sum_{C^{\prime }}\omega_{ij}\left(C^{\prime },B\right)$ the conditional probability of observing *C* in *i* given *B* in *j*. Note that this conditional probability is able to implement epistatic interaction between sites, in difference to the independent-site approximation. In turn, Felsenstein’s recursion relation becomes
(12)$$L_{ij}^{n}\left(A,B\right)=\prod_{m\in C(n)}\left(\sum_{C,D\in A}P\left(C,D\;|\;A,B,\Delta t^{m}\right)L_{ij}^{m}\left(C,D\right)\right).$$
The summation over all possible configurations of two sites and the computation of the likelihood for all pairs now results in a still feasible complexity of $O(L^{2}\times M\times q^{2})$.

Of course, a naive application of this method poses a major consistency problem: two pairs sharing one residue cannot evolve independently. As a result, the inference of the most likely pairwise statistics $\omega_{ij}\left(A,B\right)$ for each pair will give globally inconsistent results. For three pairwise distinct residues—*i*, *j*, and *k*—one will typically find
(13)$$\sum_{B\in A}\omega_{ij}\left(A,B\right)\neq \sum_{C\in A}\omega_{ik}\left(A,C\right)\;,$$
i.e., marginal distributions for site *i* do not coincide when extracted from distinct pairs containing *i*. To settle this inconsistency, we propose a constrained optimization of the pairwise likelihoods over the probabilities $\omega_{ij}$, subject to the constraint that its single-site marginals equal the single-site distributions obtained using the independent-site approximation scheme developed in the previous subsection (superscript “is”). In other words, for all *i* and *j*, the following condition is imposed,
(14)$$\sum_{B\in A}\omega_{ij}\left(A,B\right)=\omega_{i}^{is}\left(A\right)\;\mathrm{and}\;\sum_{A\in A}\omega_{ij}\left(A,B\right)=\omega_{j}^{is}\left(B\right)\;,$$
where $\omega_{i}^{is}\left(A\right)$ stands for the result of the scheme described in Section 2.1.

The hope is that by extending the phylogenetic inference beyond a sitewise description, the background pairwise statistics of the evolutionary process might be recovered, therefore improving the inference of the DCA coupling parameters.

### 2.3. Optimization: Maximizing the Likelihood

The independent-site or independent-pair approximations allow for a computationally efficient estimation of the likelihood. To correct empirical frequencies *f* for phylogenetic biases, we now need to find stationary frequencies ω maximizing the approximated likelihoods: Equation (Equation 7) (Equation (Equation 12), respectively) has to be optimized over $\omega_{i}\left(A\right)$ (respectively $\omega_{ij}\left(A,B\right)$). As each site *i* or each pair (i,j) is treated independently from the others depending on the approximation used, the optimization is conducted over either *q* or $q^{2}$ parameters at a time. However, the gradient of the likelihood in both approximations is intractable, and its concavity is unknown, making the use of standard gradient ascent techniques impractical.

Here, we rely on a stochastic optimization scheme, which was empirically found to be efficient in this scenario, inspired by the work in [17]. Parameter space, i.e., the $\omega_{i}\left(a\right)$ or the $\omega_{ij}\left(a,b\right)$, is randomly sampled by making global or local random moves: in global moves, all parameters to be optimized are simultaneously changed, while in local moves only one is changed (up to subsequent normalization). The moves are only accepted if they lead to an increased likelihood. Their magnitude is decreased throughout the optimization, starting with large displacement in parameter space and ending with small adjustments. The best parameters found are returned. This scheme is rather empirical and does not guarantee convergence. However, in testing simplified scenarios where the stationary frequencies ω are known, it was found to always lead to the correct solution.

In the case of the independent-pair approximation, $\omega_{ij}\left(A,B\right)$ needs to be optimized under the constraints defined in Equation (Equation 14). For this reason, moves proposed by the stochastic exploration of parameter space need to satisfy the constraints at all times. Here, we use a reparameterization trick inspired by the definition of direct information in [18]: tentative pair frequencies are written as
(15)$$\omega_{ij}\left(A,B\right)=\frac{1}{z(J,{\tilde{h}}_{i},{\tilde{h}}_{j})}\exp\left\{J\left(A,B\right)+{\tilde{h}}_{i}\left(A\right)+{\tilde{h}}_{j}\left(B\right)\right\}\;.$$
The optimization is then conducted over the coupling parameter *J*. Whenever *J* is changed, compensatory fields ${\tilde{h}}_{i}$ and ${\tilde{h}}_{j}$ are re-estimated to satisfy the marginalization constraints. In this way, optimization is conducted in the space of frequencies that do satisfy Equation (Equation 14).

### 2.4. From Corrected Frequencies to DCA Models

Our final aim is to infer DCA models of the form Equation (Equation 1), which are corrected for phylogenetic biases. In the last section, we have described an approximation scheme for correcting the single- and two-site equilibrium frequencies. These must, in a next step, be included in an inference procedure for the couplings and fields of Equation (Equation 1).

A first simple idea would be to use mean-field DCA [8], i.e., to invert the inferred covariance matrix $C_{ij}\left(A,B\right)=\omega_{ij}\left(A,B\right)-\omega_{i}\left(A\right)\omega_{j}\left(B\right)$ to obtain the coupling parameters $J_{ij}\left(A,B\right)$. However, there is a problem: even if we have constructed the $\omega_{ij}\left(A,B\right)$ carefully to obtain local coherence via fixing their single-site marginals to the $\omega_{i}\left(A\right)$ obtained applying the independent-site approximation, they are not globally coherent. In particular, the before-mentioned covariance matrix $C_{ij}\left(A,B\right)$ cannot be obtained as the data-covariance matrix of a sequence sample. This is easiest visible when observing the eigenvalue spectrum of the inferred *C*-matrix, which typically contains negative eigenvalues, while a data-covariance matrix is guaranteed to be positive semidefinite. Mean-field DCA uses positive pseudocounts for regularized inference, but this procedure would shift the negative eigenvalues of *C* towards larger values, and induce singularities in its in inverse.

The other popular implementation of DCA is using pseudo-likelihood maximization (plmDCA) to estimate the coupling and field parameters [19,20]. Although being more accurate than mean-field DCA, it does not use empirical single- and two-site frequencies as inputs, but the full-length sequences of the input MSA itself. To use plmDCA, we designed a way to construct an artificial MSA, which has approximately a given pairwise target statistics $\omega_{ij}^{target}$, using a simulated annealing strategy based on the work in [21]. In a first step, we emit an MSA having the correct target profile $\omega_{i}^{target}$, i.e., each column is generated independently as a sample of $\omega_{i}^{target}$. In a second step, entries inside columns are permuted in a way to establish also the target correlations contained in $\omega_{ij}^{target}$, while conserving the single-site profile $\omega_{i}^{target}$: in each move *t*, a column *i* and two rows *m* and *n* are chosen at random, and an attempt to exchange $A_{i}^{m}$ and $A_{i}^{n}$ is made. The probability of the exchange to take place is given by the Metropolis rule:(16)$$P\left(exchange\right)=\min\left[1,\exp\left(-\beta ||C^{t+1}-C^{target}\left|\right|+\beta ||C^{t}-C^{target}\left|\right|\right)\right],$$
where $C^{t}$ and $C^{t+1}$ are the covariance matrices of the current MSA before and after the exchange, and $C^{target}$ the covariance matrix corresponding to the target frequencies. ||·|| stands for the Frobenius norm of matrices, and β is a formal inverse-temperature parameter. Thus, a move is more likely to be accepted if it makes the connected correlation matrix of the alignment closer to that of the target. Parameter β is initialized at a low value and slowly increased as more moves are made. In this way, when β goes to infinity, we hope to have *C* approaching $C^{target}$ as much as possible (remember that our target $C^{target}$ cannot be reached by *C*, as only the latter is positive semidefinite).

This procedure allows us to construct a sample approximating the corrected pairwise frequencies $\omega_{ij}$, using the independent-pair approximation described above: the target frequencies are simply set to the ones resulting from the optimization of the likelihood, $\omega_{ij}^{target}=\omega_{ij}$. However, this is not possible when using the independent-site correction, since only the single site frequencies $\omega_{i}$ are corrected. In this case, we build an artificial pairwise frequency matrix defined by
(17)$$\omega_{ij}\left(A,B\right)=f_{ij}\left(A,B\right)-f_{i}\left(A\right)f_{j}\left(B\right)+\omega_{i}\left(A\right)\omega_{j}\left(B\right),$$
where $f_{i}\left(A\right)$ is the fraction of sequences in the MSA having amino acid *A* in position *i*, and $f_{ij}\left(A,B\right)$ is the fraction of sequences having simultaneously amino acids *A* and *B* in positions *i* and *j*. The pairwise statistics defined in this way has the corrected single-site frequencies as marginals, but uncorrected connected correlations. However, a major drawback of this method is that this manner of combining different frequencies gives rise to inconsistencies, with some terms $\omega_{ij}\left(a,b\right)$ being larger than 1 or smaller than 0. It is therefore impossible for our simulated annealing procedure to construct an alignment exactly reproducing these frequencies.

Once the corrected pairwise statistics are computed following Section 2, and a corresponding MSA is built, standard plmDCA is used to infer the Potts model (Equation 1).

## 3. Results

### 3.1. Design of a Toy Model

To test the methodology, we first try our methods on a toy model. This allows us to fully control the data generation, and the true model is known. As the aim of correcting data for phylogenetic bias is ultimately to have a better DCA inference, we choose our toy model to be of the Potts form. In this manner we know that using a sufficiently large i.i.d. sample the model parameters $J_{ij}$ and $h_{i}$ can be recovered with high accuracy.

For computational efficiency, the length of the model is restricted to L=25, with q=4 states for its variables. Couplings and fields are drawn from a normal distribution, with couplings taking a predominantly ferromagnetic form:(18)$$J_{ij}^{0}\left(a,b\right)=s_{ij}x_{ij}^{J}\cdot \delta_{a,b}\;\mathrm{and}\;h_{i}^{0}\left(a\right)=x_{i}^{h}\left(a\right),$$
where $\left\{x_{ij}^{J}\right\},i,j\in \left\{1\ldots L\right\}$ and $\{x_{i}^{h}\left(a\right)\},i\in \left\{1\ldots L\right\},a\in \left\{1\ldots q\right\}$ are Gaussian random variables:(19)$$x_{ij}^{J}∼N\left(\mu_{J},\sigma_{J}\right)\;\;\mathrm{and}\;\;x_{i}^{h}∼N\left(\mu_{h},\sigma_{h}\right)$$
with $\mu_{J}=0.8,\sigma_{J}=0.2,\mu_{h}=0$, and $\sigma_{h}=0.6$. The $s_{ij}$ are discrete binary variables taking values in {0,1}:(20)$$s_{ij}=\left\{\begin{array}{cc} 1 & \mathrm{with}\;\mathrm{probability}\;c/L,\\ 0 & \mathrm{with}\;\mathrm{probability}\;1-c/L. \end{array}\right.$$
To mimic the effect of structural contacts, we dilute the couplings by taking a value of c=3, making the graph underlying the coupling matrix a sparse random graph [22]: each site *i* shares a direct coupling $J_{ij}$ with 3 other sites *j* on average.

The corresponding “true” model will be called $P^{0}\left(\underline{A}\right)$ in the following, it will constitute the ground truth, against which our inference results can be tested.

### 3.2. Artificial Data

To simulate the effect of phylogeny, we sample the toy model $P^{0}$ using MCMC (Markov Chain Monte Carlo) simulations on a binary tree: Each branch of the tree corresponds to an independent finite-time MCMC run. For a branch of length Δt, a number of “mutations” is drawn from a Poisson distribution with mean μLΔt, with μ being the mutation rate per site and time unit. For each of these mutations, a site *i* is chosen at random and its new state is drawn from the local conditional probability $P^{0}\left(A_{i}|A_{1},...,A_{i-1},A_{i+1},\ldots ,A_{L}\right)$ in a Gibbs-sampling manner.

To generate an MSA, first, a root configuration is drawn from $P^{0}$, duplicated onto the two outgoing branches, and the described finite-time MCMC runs are performed. This process is iterated, taking the two resulting configurations as new roots, thus growing the tree. For *K* iterations, the resulting tree will consequently have $2^{K}$ leaves, whose Potts configurations are reported as artificial MSA.

This scheme guarantees that the number of mutational events will correspond to dynamical models in Equations (Equation 8) and (Equation 11). However, the way residues are re-drawn after a mutation depends on the full current sequence through distribution $P^{0}$, unlike the simplifying assumptions of the propagators.

For simplicity reasons, μLΔt is set to be identical for all branches of the tree, taking values 3, 5 or ∞ (i.e., μΔt≫1), resulting in respectively strong, weak and absent phylogenetic effects. In the following, the samples corresponding to finite values of Δt will be referred to as biased samples, whereas the one corresponding to Δt→∞ will be referred to as a fair or i.i.d. sample. 12 duplication events are performed, resulting in a tree of $2^{12}=4096$ leaves and $2^{12}-1$ internal nodes. Finally, so as not to depend on the particular choice of the root configuration, 30 repetitions of the sampling process are performed for each Δt.

To keep the main text concise, only results concerning the μLΔt=3 are shown. This represents the hardest case, as phylogeny effects are more pronounced for short branch lengths. Results for μLΔt=5 are shown in the Appendix A in the form of figures.

Note that, for a model without couplings, the data generating process would correspond exactly to the dynamics described in Section 2.1. For a coupled model, however, the real μ may differ from the one to be used to fit our independent-site or -pair models, due to a slowing down of the MCMC dynamics. We, therefore, use the strategy described in Section 2.1: For each sequence pair, the Hamming distance and the evolutionary time separation are calculated. Times are binned (in the simplified data generation times are actually discrete), and average Hamming distances are computed. The resulting data are fitted against the theoretical result in Equation (Equation 10) to obtain the effective mutational parameter to be used in the phylogenetic inference. Results for μLΔt=3 are shown in Appendix A, choosing Δt=0.3 without loss of generality.

### 3.3. Phylogenetic Inference Corrects the One- and Two-Point Statistics

To assess the quality of the phylogenetic correction, we first compare single-site and pairwise statistics before and after our inference to the same observables measured in an i.i.d. sample drawn from $P^{0}$.

In the case of the independent-site approximation, the single-site statistics are corrected. Observables measured in the biased sample, i.e., the sample coming from the leaves of the tree, without correction, referred to as the “tree” sample, will be denoted as $f_{i}^{t}$. After phylogenetic correction, we call the single-site frequencies $f_{i}^{inf}$. The statistics of the i.i.d. sample is $f_{i}^{0}$, obviously without any correction applied.

As demonstrated in Figure 3, the inference clearly improves the estimation of single-site frequencies over naive counting in the biased sample. Pearson correlations between $f_{i}^{inf}$ and $f_{i}^{0}$ are significantly higher than between $f_{i}^{t}$ and $f_{i}^{0}$, being larger than 0.75 in 27 out of 30 repetitions. This contrasts with the remarkably low correlations of 0.4 that can be achieved for some realizations of the tree if no correction is performed. Similarly, the slope of a linear regression of $f_{i}^{inf}$ against $f_{i}^{0}$ tends to be much closer to 1 in most cases, also showing lower variation from repetition to repetition.

A similar comparison is made for pairwise frequencies in the case of the independent-pair approximation. We now compare $f_{ij}^{t}$ and $f_{ij}^{inf}$ to their counterpart from the i.i.d. sample $f_{ij}^{0}$. The two top panels of Figure 4 once again show an improvement resulting from the phylogenetic inference, as pairwise statistics are closer to match $f_{ij}^{0}$ after it is performed.

However, one has to keep in mind that some of this improvement is due to the single-site correction. Indeed, in the independent-pair approximation, marginals of the pairwise frequencies are constrained to match the corrected single site frequencies $f_{i}^{inf}$. To evaluate the intrinsic quality of the pairwise method, we focus on the connected correlations $c_{ij}=f_{ij}-f_{i}f_{j}$, thus removing the influence of the single-site correction. Bottom panels of Figure 4 demonstrate that even this intrinsically pairwise quantity is recovered with higher accuracy after inference, even if to a somewhat lesser extent than for the frequencies. Even our very crude approximation—considering every pair as evolving independently—can correct some of the statistical bias due to phylogeny, improving over naive counting in the MSA.

### 3.4. DCA Parameters are Recovered with Increased Accuracy

We infer DCA models based both on the uncorrected and the corrected frequencies $f_{ij}^{t}$ and $f_{ij}^{inf}$ using the methodology described in Section 2.4. To evaluate both of our approximations, we infer the DCA model in the case of the single-site correction and the independent-pair correction.

In the top panel of Figure 5, inferred parameters are compared to the true ones $J^{0}$ and $h^{0}$ using Pearson correlation as a measure. Both methods—single sites and independent pairs, labeled as pairwise in the figures—lead to a significant improvement in the inference of fields. However, the inference of couplings is deteriorated when using only the single site correction, whereas it is improved in the pairwise case. This may be due to the inconsistencies appearing when combining correlations from the biased sample with corrected single site frequencies, as is explained in Section 2.4. Indeed, such inconsistencies (frequencies larger than 1 or smaller than 0) were observed for all of the 30 repetitions.

To understand if an inferred DCA model $\hat{P}$ is a good fit to the true distribution, we compute its symmetrized Kullback–Leibler (KL) divergence to the data-generating model $P^{0}$:(21)$$D_{KL}(\hat{P}\left|\right|P^{0})+D_{KL}(P^{0}\left|\right|\hat{P})={\langle H_{\hat{P}}-H_{P^{0}}\rangle }_{P^{0}}+{\langle H_{P^{0}}-H_{\hat{P}}\rangle }_{\hat{P}}\;,$$
where $H_{P}$ indicates the Hamiltonian (or, up to an additive and sequence-independent term, log-probability) of a statistical model *P*, and ${\langle \cdot \rangle }_{P}$ the average over *P*. Although the standard $D_{KL}$ depends on the intractable calculation of the partition function of one of the distributions, its symmetrized version can be easily estimated by MCMC sampling from the average energies of the two models, evaluated on samples of each model. It is a reasonable information theoretic distance measure for distributions, as it is zero if and only if the two distributions coincide, and positive otherwise. Figure 5B shows a histogram of this quantity for the 30 repetitions of the sampling process. A clear ranking between methods appears, with the inference based on the biased sample being the worst. Both phylogenetic corrections result in a model that is closer to $P^{0}$, with anadvantage for the pairwise method. Surprisingly, the decrease in inference quality of the couplings when using the single-site correction does not appear to have a strong influence on Kullback–Leibler divergence, as there is a very large drop of this quantity between a biased sample or a single-site correction based DCA.

Note also that the imperfect nature of our approximation scheme becomes visible in the figure: the KL divergence of the model inferred from an i.i.d. sample can be seen as a lower bound for what can beobtained with a finite sample. It is substantially smaller than even the pairwise correction using the same sample size.

Another important test of the model quality, in particular for protein systems, is the “contact prediction”: strong couplings between pairs of sites are expected to correspond to the sparse graphical structure of the model $P^{0}$ used for data generation. To this end, couplings are ordered with respect to their coupling strength (measured by the Frobenius norm of the coupling matrix in so-called zero-sum gauge, cf. [20]); the positive predictive value (PPV) is the fraction of true predictions (nodes connected by a link in the ground truth) in between the *N* first predictions. It is plotted as function of *N* in Figure 5C. The inference based on the i.i.d. sample is perfect in this case, ranking couplings on true links before those being not adjacent in the ground truth. The inference based on the biased “tree” sample is performing slightly worse, and it is partially corrected by the pairwise correction. On the contrary, as can already be expected from Figure 5A, the single-site correction deteriorates the reconstruction of the interaction graph.

### 3.5. Improvement in the Prediction of Single Mutant’s Energies

One of the most promising application of DCA-like methods is their ability to infer the effect of mutations in proteins from the MSA of diverged homologs [23,24,25,26,27]. Here, we investigate the potential of our phylogenetic correction to enhance the accuracy of these predictions. To recreate this setting in our toy model, we consider single-site “mutants” of “wild type” artificial sequences. Wild types can be taken either in the phylogenetically biased sample, as would be the case in standard DCA, or in the i.i.d. sample, i.e., without phylogenetic correlation to the sequences in the MSA. For any wild type sequence $\underline{A}=\left(A_{1},\ldots ,A_{L}\right)$, the L×(q−1) single mutants (i.e., single-spin flips) are denoted by ${\underline{A}}_{\left(i,\alpha \right)}$, with $i\in \left\{1,\ldots ,L\right\},\;\alpha \in A\setminus A_{i}$. For each of these, the effect of the mutation is defined by the energy difference between wild type $\underline{A}$ and mutant ${\underline{A}}_{\left(i,\alpha \right)}$:(22)$$\Delta H_{i\alpha }=H\left({\underline{A}}_{\left(i,\alpha \right)}\right)-H\left(\underline{A}\right)\;.$$
H can be either the true Hamiltonian $H^{0}$ of the generative model $P^{0}$, then defining the true mutational effect, or an inferred one, corresponding to the predicted mutational effect.

To evaluate the influence of both the phylogenetic correction and the DCA methodology on the quality of predictions, we choose to also infer a profile model as a comparison point. Profile models have vanishing couplings and reproduce the single site statistics $f_{i}\left(A\right)∼e^{h_{i}\left(A\right)}$ using local fields only, different sites are independent. They have been used with success for predicting mutational effects in proteins based on the conservation profile of the MSA [28,29], and they are the asymptotic stationary distributions of the independent-site evolution model of Section 2.1.

We first focus on the single-site phylogenetic corrections. Given a model (profile/DCA), a statistics (tree/corrected), and a specific wild type sequence $\underline{A}$, we compute the Pearson correlation between the predicted energy shifts $\left\{\Delta H_{\left(i,\alpha \right)}\;|\;i\in \left\{1,\ldots ,L\right\},\;\alpha \in A\setminus A_{i}\right\}$ and the true ones, $\left\{\Delta H_{\left(i,\alpha \right)}^{0}\right\}$. This is done for all sequences either in the biased or the i.i.d. sample, and resulting correlations are averaged over each sample. The resulting value represents thus the quality of predictions of the energies of single mutants with wild types in a given sample.

As is shown in Figure 6, when the reference sequence is taken in the biased sample, all methods seem to perform equally well, apart from the profile model inferred on the biased frequencies. In particular, applying the DCA methodology and thus attempting to fit correlations or using a simple profile model on corrected data seems to result in the same improvement.

The picture changes when the reference sequence is taken in a fair sample, i.e., when it is independent from the sample used for model inference. In this case, the performance of both DCA on uncorrected data and of the profile models drop significantly, whereas DCA inferred on corrected frequencies remains accurate. To investigate this further, we compute the average Pearson correlation as a function of the Hamming distance of the wild type to the closest sequence in the biased sample. Appendix A shows that while the performance of the uncorrected DCA and the profile models declines rapidly when using a reference sequence far away from the biased sample, the corrected DCA has a more stable performance before large Hamming distances are reached.

As the combination of DCA and of the single site phylogenetic correction outperforms profile models or a naive DCA approach, we now consider inferring the Potts model based on the corrected pairwise frequencies. The same scoring as above is used, using all single mutants for wild type sequences in both samples and computing the average Pearson correlation across wild types. Figure 7 compares the predictions of the DCA models using the tree levels of phylogenetic correction: none, sitewise and pairwise. The latter leads to a significant improvement in accuracy of predictions, outperforming the two other methods. This stands both in the case of a wild type belonging to the biased sample or to the fair sample.

Again, we investigate the dependence of those predictions on the distance of the wild type to the closest sequence in the biased sample. The largest increase in Pearson correlation resulting from the pairwise phylogenetic inference once again happens for sequences that are far from the biased sample (Appendix A). Removing part of the phylogenetic bias seems to have a stronger influence when considering the energy landscape around sequences that are far away from the leaves of the phylogenetic tree. When using those leaves as a sample without accounting for their non-independence, the resulting model seems not to learn much about the energy landscape far away from those points. However, correcting for non-independence, even in a rather crude way, leads to a much better inference in this regard.

### 3.6. Results on Protein Data

The main application of DCA-like methods so far has been their ability to predict contacts in the three-dimensional protein structure. Strong couplings between two sites in the Potts model are a good indication of the corresponding amino acids being in contact in the protein fold. As, in the case of artificial data, couplings are inferred more accurately when frequencies are corrected for phylogeny (Figure 5), it is natural to ask whether this translates to improved contact predictions for actual protein data.

To assess the performance of our correction scheme on actual protein data, we evaluated the PPV of DCA contact predictions on five protein families(cf. Appendix A for details). Those families were chosen from the families used in [8] on the basis of having short enough sequences for our pairwise phylogenetic correction to be tractable in reasonable time, and to have potentially stronger phylogenetic correlations than current Pfam data, which are based on representative proteomes, i.e., which have undergone already some phylogeny-based sequence pruning. In contrast to the artificial data, the phylogenetic tree is not a priori known, and we have applied FastTree [30,31] for each family for tree inference. Next, for each family, three DCA models were inferred: a “naive” model based on completely uncorrected statistics; a model based on frequencies corrected by the reweighting scheme, which is the one used in common DCA implementations; and a model based on frequencies corrected by our pairwise phylogenetic inference scheme. Contact prediction was done using the standard procedure of plmDCA [20].

Figure 8 shows representative results for two of the five families. In the case of PF00013, our phylogenetic correction clearly performs worse than both reweighted and uncorrected DCA for the first 100 predictions. Note that the reweighting method does not lead to any improvement either, suggesting that the phylogenetic bias may be weak for this family, and the potential benefit of the correction is overcome by problems due to the approximations used. The picture changes for PF00046, where both correction methods—reweighting and phylogenetic inference—improve significantly over the uncorrected DCA model. Reweighting outperforms our method for the prediction corresponding to the strongest coupling, having a fraction of true predictions of ∼0.7 versus ∼0.5 for the first ten predictions. However, for a large number of predictions, the phylogenetically informed DCA model tends to have an enriched fraction of contacts among its couplings when compared to the reweighted model. This observation fits well with results on artificial data, showing an overall increase in the accuracy of inferred couplings. However, as applications of DCA usually rely on the very strong couplings only, this long-term increase in accuracy remains of limited practical interest.

Results for three other families can be seen in Appendix A. Over all the five investigated protein families, our phylogenetic correction only shows improvement with respect to an uncorrected model for two of them: PF00046 and PF00111. In both cases, it is outperformed by reweighting in the first predictions.

## 4. Discussion

In this paper, we propose a principled way to correct for phylogenetic effects in the inference of Potts models from sequence data. Although the standard technique to account for these effects in coevolutionary analyses relies on an empirical reweighting of sequences, our method aims at doing so using the phylogenetic tree as well as an evolutionary model. The global nature of Potts models implies that the evolutionary model used should depend on the full sequence. However, such a global approach is intractable in the case of discrete variables such as amino acids. To overcome this problem, we proposed two subsequent levels of approximation: the first one relying on sites evolving independently as in standard models of sequence evolution; the second one describing pairs of sites, which display internal correlations but evolve independently from the rest of the sequence.

We show that our phylogenetic correction method combined with these approximations is efficient in the case of artificial data. When data are generated by a known Potts model using a sensible but simple evolutionary process on a known tree, our method is able to efficiently correct single-site and pairwise statistics, including connected correlations as intrinsically pairwise quantities. This, in turn, results in an improved inference of the Potts model in all tested aspects: individual coupling and field parameters are more accurate, the inferred Potts probability distribution is closer to the real one, contact prediction is more precise, and prediction of local energy changes from mutations is improved.

In the case of actual protein families however, results are at best mitigated. For only two of the five investigated families (PF00046 and PF00111), our method does improve the accuracy of contact predictions with respect to uncorrected data, whereas it has a negative effect on these predictions for two of the other families (PF00013 and PF00014). Furthermore, even in the positive cases, it is still outperformed by the simpler empirical method of reweighting sequences according to simple sequence-similarity measures.

In this regard, it is important to note two things: The first is that for the two families for which our method fails, the reweighting technique leads to very marginal improvements in terms of contact prediction. This seems to indicate that our method does perform reasonably well only in the case of strong phylogenetic biases. It also suggests that phylogeny does not affect contact prediction to a noticeable degree in some families. The second is that in the cases of protein families for which our method does provide an improvement, it outperforms reweighting in the “long run”, e.g., for more than ∼100 predictions for PF00046. This may mean that phylogeny has a strong effect on weaker DCA couplings that reweighting fails to correct. Even though these are not necessarily relevant for contact prediction, they impact the accuracy of the model in other aspects, such as predicting mutational effects or generating new sequences. If one wants to use DCA as a sequence model rather than simply a contact-prediction tool, it becomes all the more important to correct for phylogeny if it has a global influence on all parameters of the model. If this is the case, it is arguable that principled methods such as ours would be more appropriate than uninformed methods such as reweighting at correcting subtle effects of phylogeny.

Different reasons can be invoked for the mitigated results on protein families. One is that our method relies on the exact knowledge of the phylogenetic tree, depending both on its topology and on branch lengths. This knowledge is of course not available for proteins, where we rely on inference software to find a tree. Inaccuracies in this tree inevitably affect our method in a negative way. Another possible problem is the stationary and Markovian nature of our evolutionary model, which may not be true in the case of protein evolution. Over evolutionary time scales, variable environments lead to changing selective pressures, population sizes and mutation rates, which are currently not accounted for by our model. However, we expect that the major problem lies in the nature of the approximations we had to resort to. The first one, independent sites, is in contradiction with the global nature of the Potts model we try to infer. The second—independent pairs—allows for the correction of pairwise statistics, but suffers from obvious consistency problems since overlapping pairs of sites cannot be considered independent. Note that this has an important consequence, when we go to protein families with longer sequences: whereas phylogenetic tree inference becomes more accurate for longer sequences, and the independent pair approximation requires $O(L^{2})$ inferences for all pairs of residue positions, with *L* being the sequence length. As a consequence, the before-mentioned inconsistencies are expected to grow drastically with sequence length.

The necessity for these approximations comes from two characteristics of the class of models we are using: their global nature, in the sense that they give probabilities to sequences in a nonfactorized way, and the discrete nature of the variables used (amino acids). By rendering certain calculations intractable, such as tracing over all possible states of internal nodes in the tree; these two characteristics make the use of approximations unavoidable. In this article, approximations attempt to circumvent the global nature of the Potts model by factorizing probabilities in different ways, namely, sitewise and pairwise. However, an interesting different class of approximations would be to forget about the discrete nature of amino acids and model them by continuous variables instead. This would transform the Potts model into a Gaussian distribution, making the design of a global propagator tractable. Note that on similar grounds gaussDCA [32], an analytically solvable Gaussian version of DCA, was developed a few years back, and it was found to perform similar to other DCA techniques in contact prediction.

Another interesting alternative might be built upon the observation made in [14]: phylogenetic correlations between the sequences of the training MSA lead to a fat tail of large eigenvalues of the covariance matrix, i.e., of the empirically observed statistics reproduced by DCA models. Furthermore, it was argued in [33] that the corresponding eigenvectors are extended over many positions and amino acids, thereby giving rise to many small couplings. The contact prediction was found to be more closely related to small eigenvalues of the covariance matrix, with localized eigenvectors giving rise to large localized couplings. However, while phylogenetic correlations between sequences are sufficient to generate extended eigenvectors with large eigenvalues, the latter may also result from slightly different functionalities of subfamilies of the studied MSA, i.e., they may contain biologically sensible information, cf. [34,35]. Disentangling the two—sequence clustering by phylogeny and by subfunctionalization—seems a nontrivial task.

As DCA-like pairwise models are increasingly used in sequence analysis, and as their ability to accurately model sequence variability in protein families gets more established, the need to infer parameters more accurately and without bias increases. For this reason, correcting for phylogeny in a controlled and principled way is essential. Whether this can be achieved using techniques similar to the one presented in this paper, or using different types of approximations as the two mentioned in the last to paragraphs, or totally different techniques, remains a widely open and challenging question.

## Figures and Tables

**Figure 1 entropy-21-01090-f001:**
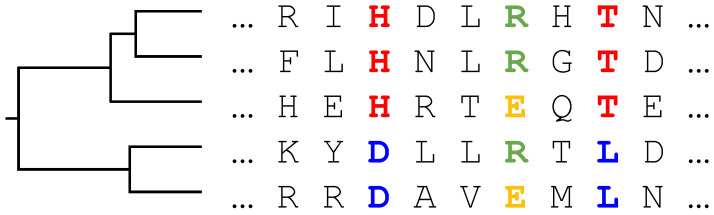
Homologous proteins constituting a multiple-sequence alignment (MSA) are related by common ancestors through a phylogenetic tree.

**Figure 2 entropy-21-01090-f002:**
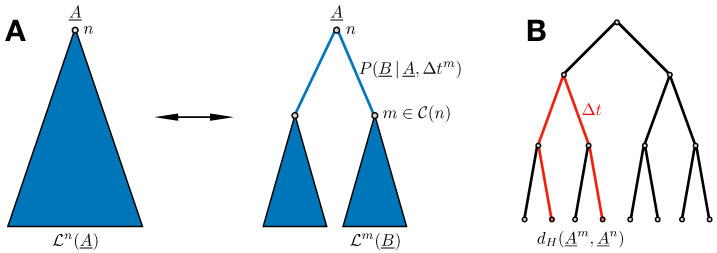
(**A**) Illustration of Equation (Equation 5): $L^{n}\left(\underline{A}\right)$, as represented on the left, is the probability of observing all sequences in the MSA having node *n* as common ancestor, given the sequence $\underline{A}$ of this ancestor. This probability can be decomposed into a product over contributions of node *n*’s children m∈C(m). For each child *m*, we have to consider the propagator $P(\underline{B}\;|\;\underline{A},\Delta t^{m})$ from *n* to *m*, times the probability $L^{m}\left(\underline{B}\right)$ associated with the subtree rooted in *m*, and summed over all possible configurations $\underline{B}$ of *m*. Note that the sum over each child can be done independently; therefore, Felsenstein’s algorithm runs in linear time in the number of internal nodes. (**B**) Measuring Hamming distances and time separations between sequences: thanks to the stationary dynamics of Felsentein’s model, the time-dependence of the Hamming distance between a parental and a child configuration can be estimated from observed leaf configurations. To this end, for any two leaves, ${\underline{A}}^{m}$ and ${\underline{A}}^{n}$, we determine the Hamming distance $d_{H}\left({\underline{A}}^{m},{\underline{A}}^{n}\right)$ and the time separation Δt, the latter by summing the lengths of all branches on the connecting path. Time binning and averaging are used to estimate the curve ${\bar{d}}_{H}\left(\Delta t\right)$.

**Figure 3 entropy-21-01090-f003:**
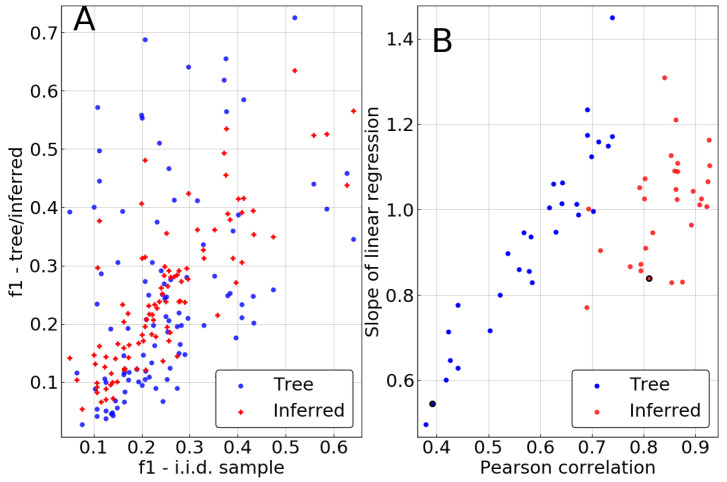
Result of the single-site phylogenetic inference for μLΔt=3. (**A**) Single-site statistics of a sample of $P^{0}$ coming from a tree, before (“Tree”), and after (“Inferred”) phylogenetic inference, against “true” single site statistics coming from the fair i.i.d. sample. (**B**) Slope of the linear regression and Pearson correlation corresponding to the plot in panel (A), for the 30 repetitions of the experiment. The black-circled points correspond to the sample displayed in panel (A).

**Figure 4 entropy-21-01090-f004:**
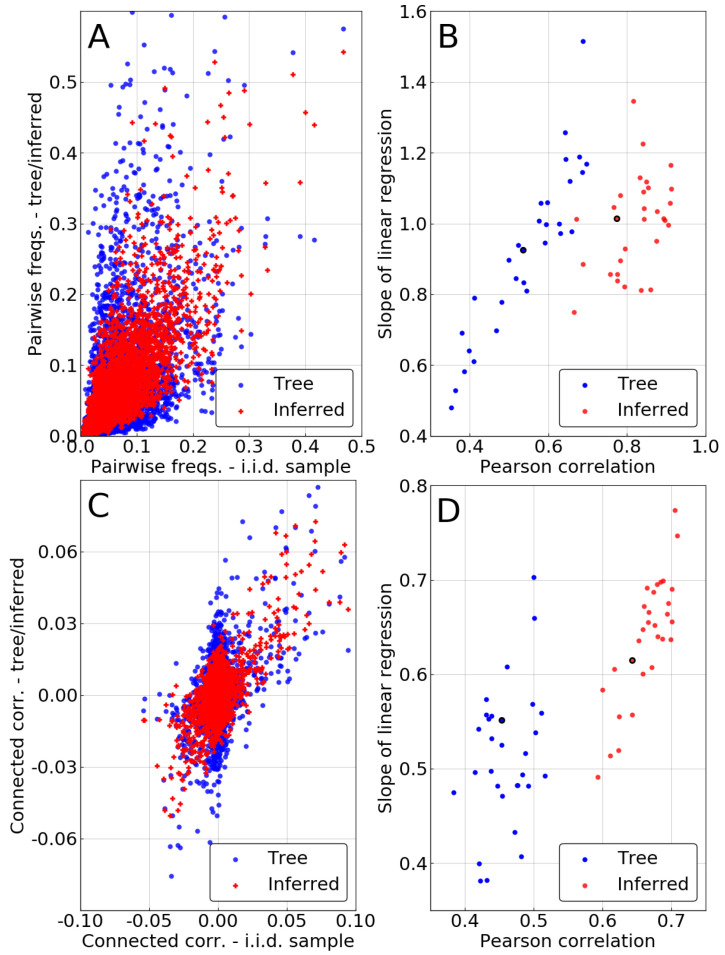
Result of the pairwise phylogenetic inference for μLΔt=3. (**A**) Pairwise frequencies $f_{ij}\left(a,b\right)$ of a sample of $P^{0}$ coming from a tree, before (“Tree”), and after (“Inferred”) the phylogenetic inference, against “true” pairwise frequencies coming from the fair sample. (**B**) Slope of the linear regression and Pearson correlation corresponding to the plot in panel (A), for the 30 repetitions of the experiment. The black-circled points correspond to the repetition displayed in panel (A). (**C**) Same as panel (A) for connected correlations $c_{ij}=f_{ij}-f_{i}f_{j}$. (**D**) Same as panel (B) for connected correlations.

**Figure 5 entropy-21-01090-f005:**
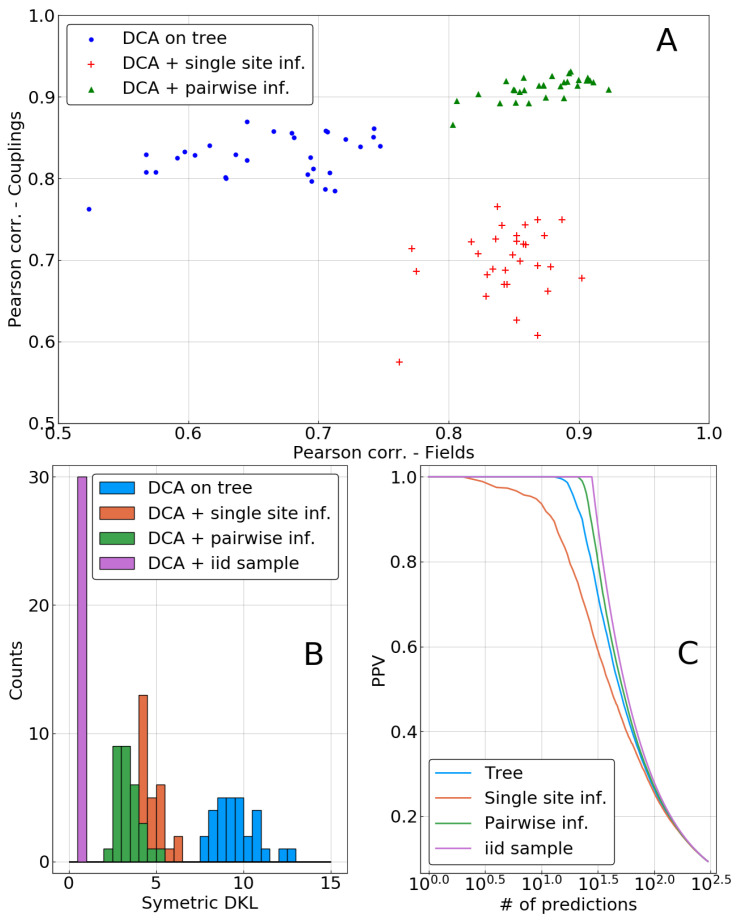
Direct coupling analysis (DCA) models inferred after single-site or pairwise phylogenetic correction for μLΔt=3. (**A**) Pearson correlation between parameters of inferred and of true DCA models. *y*-axis: couplings $J_{ij}$; *x*-axis: fields $h_{i}$. One point corresponds to one repetition of the MCMC process on the tree, i.e., to one sample. (**B**) Histogram of the symmetrized Kullback–Leibler divergences between inferred and true models for all samples. (**C**) Positive predictive value for predicting non-zero couplings (i.e., “contacts”) using inferred DCA models. DCA inferred on the i.i.d. sample performs perfectly in this case.

**Figure 6 entropy-21-01090-f006:**
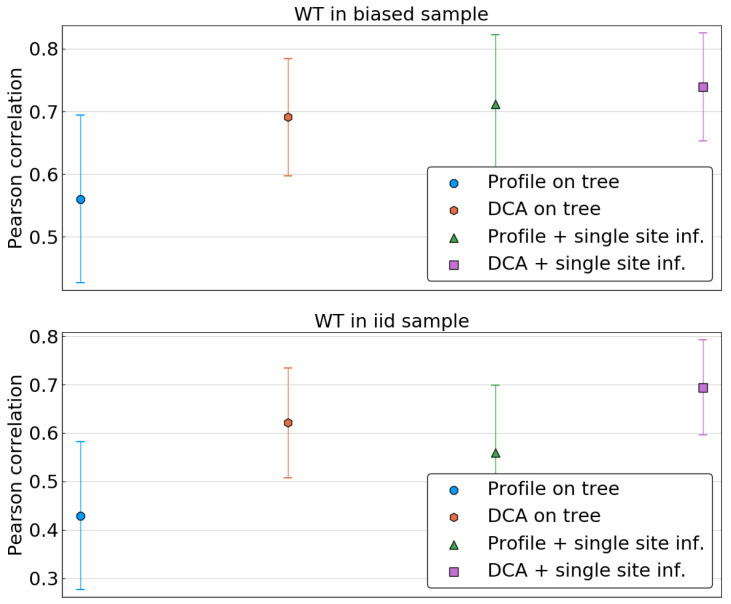
Pearson correlation in predicting energies of single mutants averaged over sets of reference sequences for μLΔt=3. In the top panel, reference sequences are taken in the biased sample, i.e., among the leaves of the phylogenetic tree. In the bottom panel, reference sequences are taken in a fair sample of $P^{0}$. Predictions are made using four models: a profile model and a Potts model trained on the uncorrected biased sample, respectively (“Profile on tree” and “DCA on tree”, respectively), and using the corrected single site frequencies (“Profile + single site inf.” and “DCA + single site inf.”, respectively). Error bars indicate the standard deviation across the 30 repetitions of the tree sampling process.

**Figure 7 entropy-21-01090-f007:**
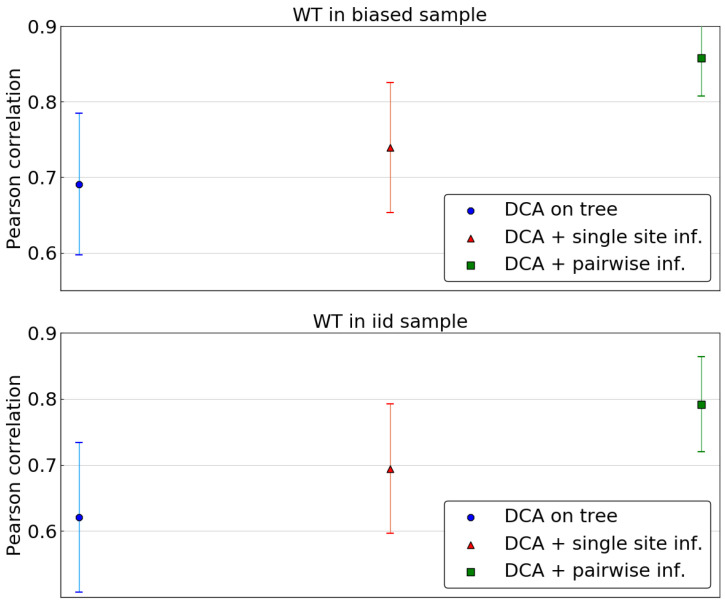
Pearson correlation in predicting energies of single mutants for μLΔt=3 averaged over sets of reference sequences. In the top panel, reference sequences are taken in the biased sample, i.e., among the leaves of the phylogenetic tree. In the bottom panel, reference sequences are taken in a fair sample of $P^{0}$. Predictions are made using a DCA model inferred either directly on biased data, either using corrected single site frequencies, either using corrected pairwise frequencies. Error bars indicate the standard deviation across the 30 repetitions of the tree sampling process.

**Figure 8 entropy-21-01090-f008:**
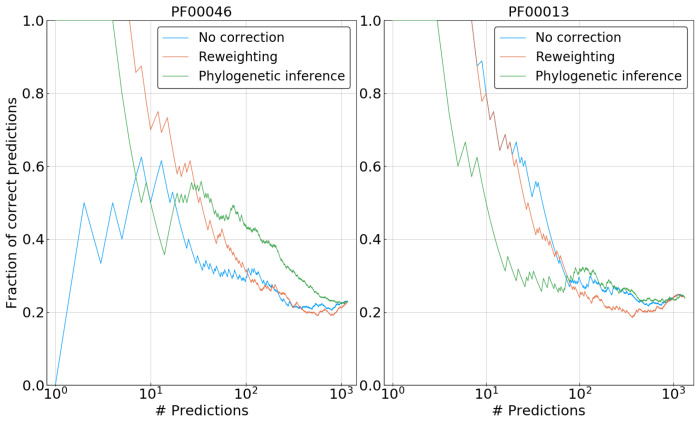
Positive predictive value for predicting contacts in representative structures for two protein families PF00013 and PF00046. The blue lines indicate a naive DCA method without any correction for phylogeny. The orange lines show results for the sequence reweighting scheme. The green lines show results after our phylogenetic inference scheme.

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
