# Peer review of "Toward Inferring Potts Models for Phylogenetically Correlated Sequence Data"

_entropy, 2019, doi:10.3390/e21111090_

Round 1

Reviewer 1 Report

The authors describe a 'principled' method to take phylogeny into account when fitting a Potts model to protein sequence alignments, in order to predict protein structure and mutation effects. The authors are honest about the limitations of their method in practice, which has mixed results in comparison to the empirical hacks that researchers currently use. While the authors' approach is interesting, it has limited practical application (as they admit). For that reason, I see this paper as presenting a valuable negative result that should caution other researchers against following this particular approach to inferring Potts models from protein sequence data.

Major comments:

1) the authors describe how their new method, in toy examples and in some protein examples, allows for more accurate inference. However, my suspicion is that any such advantages would disappear completely if computational complexity were taken into account in the authors' experiments. For instance, a cheaper algorithm that fits data to a biased model will out-perform a expensive model that fits data to the correct model, whenever the biased model can do inference on significantly more data than the expensive 'correct' model. The authors should discuss this point in their paper, and their experiments should take computational complexity as well as the maximum size of data for effective inference into account.

2) I am surprised that the authors did not more deeply discuss the meaning of the result in the recent paper by Qin Qin and Colwell Colwell (2018), nor discuss those results in the context of the excellent paper by Cocco Coccoet al. (2013) which shows that couplings and phylogeny affect distinct parts of the eigenvalue spectrum for the sample covariance matrix, especially as the senior author co-authored the Cocco et al. (2013) paper. The smallest eigenvalues reflect constraints (i.e. couplings), while the largest eigenvalues reflect global correlations such as phylogeny. This is a major point of both these papers.

3) While the authors approach is in principle "principled", a major issue is that the assumptions of the evolutionary model in practice are certainly not true. While the math is elegant, the assumptions of the principled model ignores practical realities such as variation in mutation rates, natural selection, recombination, horizontal gene transfer, variation in population size, etc. While the authors model is principled, those principles are probably violated all the time in biological reality. I was reminded by Leo Breiman's classic paper "Statistical Modeling: The two cultures" which draws a stereotype of statisticians who write down theoretical sound yet flawed generative models for real world data, in comparison to machine learning researchers who treat the data generating process as an unknown black box, and who focus solely on making good predictions. My guess is that this could, in itself, account for the authors finding that their algorithms are slower yet in most cases less accurate than re-weighting hacks, despite being more 'principled'.

Minor comments

I found numerous spelling and grammatical errors throughout. I found this very annoying, and this raises questions about the care with which the authors wrote their manuscript. The title itself has two typos and should read: "Toward inferring Potts models for evolutionarily correlated sequence data".

Despite these concerns, the science reported here is solid. However, to me the impact of this paper is how it reveals the flaws of expensive 'principled models' built on biologically questionable assumptions in comparison to practical hacks like re-weighting or black box models like deep learning that are trained on empirical data.

Reviewer 2 Report

In the paper by Horta et al, a new global evolutionary model is proposed, taking into account besides MSA aminoacid coevolution, also phylogenetic relationships. This method allows corrections for diversity biases within MSAs of protein families and it outperforms current methods such as re-weithging of similarities when applied to "ideal" datasets. With "real" protein data, the presented results are not so exciting but there might be a biological reason for that (see below). 

To my knowledge, this is the first model that incorporates phylogenetic information in co-evolutionary analysis. Thus although not perfect, it represents an important step to improve computer based predictions of e.g.  residue-residue contacts.

Although not being able to fully evaluate the mathematical model proposed, I have some suggestions that I would like the authors to consider.

I was disappointed that no information nor discussion regarding the chosen PF families was given. I am assuming sequences and alignments were retrieved from the PFAM database but this information is absent from the manuscript. Moreover, it is not clear if the MSAs contained low or high sequence biases, the number of sequences present in the MSA, the global identity of the sequences, nor how many informative sites were used for phylogenetic reconstruction. A section with this information in both methods and results/discussion sections should be added.

Also, it is known that phylogenetic reconstructions based on small proteins (thus, small alignments) generally are unreliable, and produce many low supported branches (low Bootstrap values). This is due to the small number of informative sites (non-conserved columns in the alignment) and poor phylogenetic signal.The mathematical model does not have enough information to constrain with statistical support the solution space. Although the length of the alignments is not discussed in the text, a search in PFAM database showed that the selected PFAMs correspond to small proteins with less than 100 aminoacids. I would suggest the authors, if computationally feasible, to apply the model to longer protein families or at least, to discuss the possible effect of alignment length/number of informative sites in what regards phylogenetic reconstruction and the model results when compared to other methods.  I suspect that the model would have considerable better results when applied to longer protein alignments.

Author Response

For detailed responses, please see the attachment below.

Reviewer 3 Report

Dear Editor and Authors,

The manuscript presents a problem in the study of protein family alignment (dependence on protein family samples studied) and proposes essential corrections for the study of the phylogeny. The study is very relevant to evolution studies and the arguments presented are reliable and coherent. That is why I recommend its publication.

Sincerely,

Author Response

We acknowledge the positive evaluation of our manuscript. 

Round 2

Reviewer 1 Report

The authors have done a good job revising their manuscript and I recommend it be accepted for publication in this special issue.

Note that there is a typo introduced in the last sentence ("to" should be "two").

Author Response

We thank the reviewer for reading also the revision of our paper, and are happy to see that the reviewer is satisfied by our revision.

Reviewer 2 Report

The authors addressed all my comments and I have no further concerns regarding this paper.